# PTEN Dual Lipid- and Protein-Phosphatase Function in Tumor Progression

**DOI:** 10.3390/cancers14153666

**Published:** 2022-07-28

**Authors:** Anne Liu, Yanyu Zhu, Weiping Chen, Glenn Merlino, Yanlin Yu

**Affiliations:** 1Laboratory of Cancer Biology and Genetics, National Cancer Institute, National Institutes of Health, Bethesda, MD 20892, USA; afwngosm@gmail.com (A.L.); 2157179272az@gmail.com (Y.Z.); gmerlino@helix.nih.gov (G.M.); 2National Institute of Diabetes and Digestive and Kidney Diseases, National Institutes of Health, Bethesda, MD 20892, USA; weipingchen@niddk.nih.gov

**Keywords:** PTEN, PTEN lipid phosphatase, PTEN protein phosphatase, mutation, PTEN protein substrate, tumorigenesis, metastasis

## Abstract

**Simple Summary:**

Phosphatase and tensin homolog deleted on chromosome ten (PTEN) is a multifunctional tumor suppressor with protein- and lipid-phosphatase activities. The inactivation of PTEN is commonly found in all human cancers and is correlated with tumor progression. PTEN-lipid-phosphatase activity has been well documented to dephosphorylate phosphatidylinositol-3, 4, 5-phosphate (PIP3), which hinders cell growth and survival by dampening the PI3K and AKT signaling activity. PTEN-protein-phosphatase activity is less well studied and understood. Recent studies have reported that PTEN-protein-phosphatase activity dephosphorylates the different proteins and acts in various cell functions. We here review the PTEN mutations and protein-phosphatase substrates in tumor progression. We aim to address the gap in our understanding as to how PTEN protein phosphatase contributes to its tumor-suppression functions.

**Abstract:**

PTEN is the second most highly mutated tumor suppressor in cancer, following only p53. The PTEN protein functions as a phosphatase with lipid- and protein-phosphatase activity. PTEN-lipid-phosphatase activity dephosphorylates PIP3 to form PIP2, and it then antagonizes PI3K and blocks the activation of AKT, while its protein-phosphatase activity dephosphorylates different protein substrates and plays various roles in tumorigenesis. Here, we review the PTEN mutations and protein-phosphatase substrates in tumorigenesis and metastasis. Our purpose is to clarify how PTEN protein phosphatase contributes to its tumor-suppressive functions through PI3K-independent activities.

## 1. PTEN Dual Lipid and Protein Phosphatase

In 1984, scientists discovered that the loss of part or all of chromosome 10 was associated with brain, bladder, and prostate cancer [1,2,3]. Conversely, the reintroduction of wild-type chromosome 10 into glioblastoma-cell lines reduced the ability of the tumor formation in nude mice [4], which suggested an important tumor-suppressor role for chromosome 10. Later, a chromosome loss-of-heterozygosity (LOH) analysis identified region 10q23 as the most common region of loss on chromosome 10 in prostate cancer [5], which suggested that this region contains a critical tumor-suppressor gene. Then, in 1997, several laboratories identified a putative tumor-suppressor gene at 10q23 that encodes a 403 amino acid protein with a protein-tyrosine-phosphatase domain and homology to chicken tensin and bovine auxilin, which was then named PTEN for phosphatase and tensin homolog deleted on chromosome 10 [6,7,8,9]. The PTEN protein contains five major domains: the N-terminal PIP2-binding domain (PBD), the catalytic domain of the phosphatase, the C2 domain, the C-tail domain, and the PDZ-binding domain (PDZ/BD); the C-terminal PDZ/BD domain can inter- act with other proteins [10,11,12] (Figure 1A). The PTEN phosphatase domain contains a signature CX5R motif that forms its catalytic pocket, which is also known as the P- loop. The cysteine at the base of this pocket allows the phosphatase to react with substrates. Moreover, later studies confirmed that PTEN not only specifically dephosphorylates protein substrates at tyrosine-, serine-, and threonine sites, but al- so dephosphorylates lipid phosphatidylinositol 3,4,5-trisphosphate [PtdIns(3,4,5)P3] to form phosphatidylinositol 4,5-bisphosphate [PtdIns(4,5)P2]. PTEN also contains a TI-loop that contributes to its lipid-phosphatase activity by determining the size of the catalytic pocket [10,13]. The structure of the catalytic domain of the phosphatase and C2 domains of PTEN characterizes its dual specificity. The C2 domain binds to phospholipids in a Ca2+-independent manner [10] (Figure 1A). PTEN-lipid-phosphatase activity is critical for its tumor-suppressor function, which antagonizes the PI3K–AKT/PKB signaling pathway by dephosphorylating phosphoinositides, thereby modulating the cell-cycle progression and cell survival [7]. PTEN protein phosphatase can dephosphorylate multiple substrates, playing diverse roles in cellular function, including inhibiting cell adhesion and migration and maintaining genomic integrity [12,14].

PTEN protein is mainly found in the cytoplasm. However, the study of the crystal structure of PTEN suggested that its C2 domain helps it bind to phospholip-id membranes [10]. SUMO1 modification at the K266 and K254 sites in the C2 domain promotes the cooperative binding of PTEN to the plasma membrane via electrostatic interactions, downregulating the PI3K/AKT pathway [15]. PTEN localized in the cy- toplasmic membrane performs its lipid-phosphatase activity by converting PIP3 to PIP2. PTEN protein can also shuttle to the nucleus to maintain genomic stability. By physically associating with CENP-C (centromere proteins), which are an integral component of the kinetochore, PTEN regulates RAD51 expression, which reduces the incidence of spontaneous double-stranded breaks (DSBs) and plays a role in DNA repair [14]. Nuclear PTEN also exhibits its tumor-suppressive effect through G1-phase cell-cycle arrest by preventing cyclin D1 localization and decreasing the level of cyclin D1 [16], together with the effects of PTEN on p27Kip1, to suppress the cell cycle [17,18] (Figure 2).

Recent studies have reported that PTEN protein could be exported to other cells through exosomes. Ndfip1, which is an adaptor molecule for the neural precursor cell-expressed developmentally downregulated protein 4 (NEDD4) family of E3 ubiquitin ligases, is required for the recruitment and secretion of PTEN to recipient cells. Lysine 13 within PTEN is also necessary for the exosomal transport of PTEN and ubiq1uitination by NEDD4-1 [19]. PTEN can exert phosphatase activity in the recipient cells and reduce AKT phosphorylation and cell proliferation.

PTEN has a wide variety of cellular functions, such as inducing apoptosis and G1-phase cell-cycle arrest, inhibiting cell survival, motility, adhesion, and migration, as well as maintaining genomic integrity (Figure 2). PTEN induces apoptosis via the blocking of the PI3K/AKT pathway. Under low-growth-factor cell-culture conditions, both PTEN-mediated growth suppression and PTEN-induced cell death are enhanced [20]. The C2 domain of PTEN allows it to inhibit cell migration following the auto-dephosphorylation of threonine 383 through its protein-phosphatase activity. This is separate from its lipid-phosphatase activity, which reveals the importance of the protein-phosphatase activity for PTEN as a tumor suppressor [21].

PTEN functions as a haploinsufficient tumor suppressor, where the protein produced after the deletion of one allele on chromosome 10 is insufficient for proper function [22] (Figure 3). The complete loss of PTEN will trigger senescence and often leads to early death in genetically engineered mice (GEM) [23]. Mammary epithelial cells from PTEN hypermorphic (Ptenhy/+) mice with 80% of the regular PTEN expression showed enhanced proliferation and increased resistance to apoptosis after ultraviolet irradiation [24]. The subtle reduction in the normal PTEN levels led to decreased survival and the increased activation of Akt protein, as well as increased cancer susceptibility (Figure 3). In contrast, a higher level of PTEN reduces the tumor incidence and promotes longevity [25] (Figure 3). Interestingly, PTEN protein can form homodimers [26], and the mutation of one PTEN allele has dominant-negative effects in cancer via PTEN dimers [27] (see below). 

## 2. PTEN Alternative Isoforms and Their Functions

New studies have shown that PTEN has three alternative translational isoforms: PTENα, PTENβ, and PTENε, which are produced from the same mRNA as canonical PTEN and are generated due to non-AUG translational initiation. Each has a longer N-terminal extension than the canonical PTEN protein (Figure 1B). PTENα, which is also known as PTEN-long, was the first isoform identified [28,29]. PTENα is 173 amino acids longer at the N-terminal than canonical PTEN, which is translated at the initiation site at the CUG codon in the 5’-untranslated region (UTR) of PTEN mRNA. PTENα translation is controlled by eukaryotic translation initiation factor 2A (eIF2A), and it requires a CUG-centered perfectly palindromic motif. PTENα is predominantly localized in the cytoplasm, and it functions in metabolism by inducing cytochrome c oxidase (COX) activity and ATP production in the mitochondria. In the presence of substantial levels of canonical PTEN, PTENα overexpression increases mitochondrial COX activity. Additionally, PTENα interacts with canonical PTEN to increase PINK1 and promote energy production [29].

A few years after the discovery of PTENα, the same laboratory identified PTENβ as another alternatively translated isoform of PTEN [30]. PTENβ translation is initiated from the AUU codon in the 5’UTR of PTEN mRNA and is predominantly localized in the nucleolus. PTENβ physically interacts with and dephosphorylates its downstream targets in nucleoli. By studying protein phosphatase activity deficient PTENβ mutants (Y284L and C270S), it was revealed that PTENβ is a protein phosphatase for nucleolin. PTENβ is responsible for negatively regulating rDNA transcription and cellular proliferation [30]. PTENε is another N-terminal-extended PTEN isoform. Its translation is initiated from the CUG codon in the 5’UTR of PTEN mRNA, and it is mainly localized in the cell plasma membrane. Like PTENα, PTENε is also regulated by eIF2A. The protein-phosphatase activity of PTENε physically interacts with and dephosphorylates VASP and ACTR2, which are regulators of filopodia formation and cell mobility. PTENε suppresses filopodia formation, thereby decreasing the aggressiveness of cancer cells and limiting the metastasis capacity of tumor cells [31]. 

All alternative translational isoforms contain a phosphatase domain-like canonical PTEN and have tumor-suppressive functions, but they are more unstable than canonical PTEN overall. These isoforms are more sensitive to proteasome-mediated degradation, and they have shorter half-lives than canonical PTEN. Interestingly, PTENα and PTENβ can promote carcinogenesis and tumorigenesis by physically interacting with the histone 3 lysine 4 (H3K4) presenter WDR5 and promoting H3K4 trimethylation [32]. In liver-cancer cells, the expression of PTENα/β is inconsistent with that of canonical PTEN. The E3 ligase USP9X (ubiquitin-specific peptidase 9, X-linked) positively regulates PTENα/β, where it deubiquitinates and stabilizes PTENα/β by binding directly to their amino(N)-terminal extensions (NTEs), but not for canonical PTEN. FBXW11 (F-box/WD repeat-containing protein 11) degrades PTENα/β and induces their ubiquitination by directly binding to the N146 fragment in PTENα/β. Thus, FBXW11 suppresses tumorigenesis by degrading PTENα/β, while USB9X promotes tumorigenesis by stabilizing PTENα/β [32], which suggests that PTEN alternative isoforms may interact with other binding partners to play opposing tumorigenesis functions. The discoveries of PTENα, PTENβ, and PTENε reveal the complexity of the PTEN protein family and shed light on more potential targets for cancer inhibition. 

## 3. PTEN Inactivation in Cancer Progression

PTEN is one of the most highly mutated genes in human cancers. The dysfunction of PTEN by genetic mutation or epigenetic silencing contributes to most cancer development and progression (Figure 4, Table 1) [7,8,33]. It is often associated with high-grade and metastatic potential drug resistance [34,35] and poor patient prognosis [36,37,38]. The inactivation of PTEN is caused by various mechanisms, including genetic loss, point mutation, epigenetic regulation, and posttranslational modifications. Most alterations result in the loss of or reduction in PTEN protein. 

PTEN genetic loss is involved in various tumor types, such as glioblastoma, breast ductal carcinoma, endometrial carcinoma, prostate adenocarcinoma, ovary cystadenocarcinoma, melanoma, pancreatic adenocarcinoma, colorectal cancer, etc. (Table 1). In GEM mice, PTEN deficiency leads to embryonic lethality, and even a small reduction in the PTEN levels enhances the cancer incidence [23,24,39,40]. Contrarily, the systemic overexpression of PTEN in GEM models amplifies its tumor-suppressive function and protects against tumorigenesis [41], which suggests that the precise level of PTEN expression is a critical factor for the tumor-suppressor function, and that the reduction in PTEN activity is a driving mechanism for tumor progression. The pathologic characteristics that are associated with PTEN deletion in mice are phenocopied in a wide range of human tumors with loss-of-heterozygosity (LOH) mutations. In cancer, the PI3K/PTEN/Akt pathway has been identified as one of the critical molecular axes driving tumorigenesis [42,43,44,45]. The loss of PTEN activity has been reported to be responsible for many of the phenotypes of cancers, and it affects the development of 15–70% of human cancers [35,43,44] (Table 1). 

Much has been learned about the features of PTEN point mutation through the analysis of PTEN germline mutations found in patients with Cowden disease (CD) [46]. Patients with CD, who harbor missense mutations in the PTEN phosphatase domain, are cancer-prone and develop more lesions than patients with PTEN deletion, which causes the complete loss of the PTEN function [47]. Interestingly, one CD-derived conversion in PTEN (G129E) [9], which loses lipid- but not protein-phosphatase activity, cannot induce either G1 arrest or apoptosis, but retains the ability to inhibit cell spreading and motility [48]. Another CD-derived PTEN mutant (C124S), which lacks lipid- and protein-phosphatase activity, cannot block cell spreading or migration [21,48], which suggests that PTEN phosphatase plays a distinct role, and that its protein-phosphatase activity may play an essential role in tumor-cell migration (Figure 1A).

In cancer, most PTEN point mutations are also found in its phosphatase domain, including C124S, G129E, Y138L [49], and R130 point mutations, which cause the loss of PTEN phosphatase activity [27,46,50]. As the C124S mutation, R130G is also a catalytically dead PTEN variant, where both the protein-phosphatase and lipid-phosphatase activities of wild-type PTEN are lost [46,50], whereas Y138L-mutant PTEN lacks the protein-phosphatase function but retains the lipid-phosphatase activity. The majority of PTEN somatic mutations are nonsense, frameshift, or splicing mutations, and a large proportion occurs on exon 5 [46]. R130X, R233X, and R235X are mutational hotspots. According to the TCGA database, up to 2021, a point mutation at R130 is the most frequent position across all types of cancer (11.4% of all PTEN mutations) (Figure 4).

To model the function and consequence of PTEN point mutation, two groups developed knock-in mice with the mutant alleles of PTEN G129E and C124S, identified in CD [27,51]. In an early study, Wang et al. [51] showed that the homozygous nonsense PTENΔ4–5 and missense PTENC124R and PTENG129E mutants caused embryonic lethality, which suggests that lipid-phosphatase activity or total phosphatase are required for early development. Similar to that in CD patients, the mice with a heterozygous one-mutant allele appeared with similar abnormalities but different organ-specific tumors, with varying degrees of severity and age onset between PTENC124R and PTENG129E mutant mice. Although the AKT activation was unchanged in tissues with the PTEN G129E and C124R mutant forms, the PTEN-protein level was remarkably decreased or was undetectable in PTEN C124R, which suggests that the loss of the PTEN-protein-phosphatase activity caused the PTEN instability and may regulate PTEN stability [21]. More interestingly, the female mice with PTEN C124R formed no detectable lesions at the mammary gland at 15 months. Still, more than 50% of PTEN G129E mutant female mice developed large palpable tumors, which suggests that the additional loss of PTEN-protein-phosphatase activity in PTEN C124R triggered an extensive cell-death response, which was evident in early and advanced mammary tumors [52]. Later, similar knock-in mouse models, established by Papa et al., also demonstrated that PTEN-specific mutations (PTENG129E and C124S) contribute to the variable tumor phenotypes observed in patients with Cowden [27]. Notably, this study has shown that PTEN mutants (C124S and G129E, as well as R130G) can form heterodimers with wild-type PTEN, thereby suppressing the wild-type PTEN function in a dominant-negative manner [26,27]. This is an important finding because PTEN can exist in a homodimeric complex, and mutants inhibit the normal PTEN-protein function via dominant-negative dimerization mechanisms. The results may explain why the high frequency of PTEN mutation is observed in cancer, and they confirm the role of PTEN point mutations in tumorigenesis. The different spectrums of tumor incidence (including organ-specific, severity, and age onset) between the PTEN mutants G129E and C124S or C124R indicate that the PTEN-protein-phosphatase activity may play a role in tumorigenesis through at least a partially different mechanism as PTEN-lipid-phosphatase activity. For example, the accurate role of PTEN-protein-phosphatase activity should consider its substrates in a specific tissue (see the following section). Moreover, PTEN activity can be reduced through the phosphorylation of the PTEN’s unstructured C-terminal tail. The C-terminal tail of PTEN was found to play a role in stabilizing the PTEN homodimer structure, and its phosphorylation, which is affected by the loss of PTEN-protein-phosphatase activity, interferes with stabilization [21,26,49,52].

**Table 1 cancers-14-03666-t001:** PTEN Lesions (Deletion and/or Mutation) in Sporadic Human Malignancies.

Site/Tissue	Tumor Type	Range	Average	Comment	Reference(s)
Brain	Glioblastoma	12–84%	29% (88/303)	Mostly LOH	[53,54]
Breast	Ductal carcinoma	11–55%	33% (415/1257)	Mostly LOH	[55,56,57]
Endometrium	Endometrioid carcinoma	19–82%	67% (352/529)	LOH and mutation	[58,59]
Prostate	Adenocarcinoma	12–63%	33% (88/267)	Mostly LOH	[60,61]
Ovary	Cystadenocarcinoma	9–61%	30% (33/112)	LOH and mutation	[62,63]
Skin	Melanoma	11–39%	30% (57/190)	Mostly LOH	[64,65]
Thyroid	Carcinoma (ATC)	10–41%	11% (21/196)	Mostly LOH	[66,67,68]

At the transcriptional level, PTEN can be regulated by other tumor-suppressor genes and transcription factors [12,69,70], such as positive regulators (early growth response protein 1 (EGR-1) [71], tumor suppressor p53 [72], and peroxisome proliferator-activated receptor (PPAR-γ) [73]) and negative regulators (mitogen-activated protein kinase kinase-4 (MKK4) [74], c-Jun [75], transforming growth factor-beta (TGF-β) [6], and nuclear factor kappa B (NFkB) [76,77]). Recently, miRNAs, such as miR-21 and miR-181b-1, have been reported to suppress PTEN mRNA translation through posttranscriptional regulation [78]. Promoter hypermethylation is another method of tumor-suppressor-gene inactivation and regulation for PTEN [79], and it is also involved in DNA repair. Histone methylation can also regulate the PTEN function, thereby influencing cellular processes. 

There are multiple posttranslational modifications (PTMs) for PTEN, such as phosphorylation, methylation, ubiquitination, SUMOylation, and acetylation [11] (Figure 5). The PTEN C2 domain and its C-terminal can be phosphorylated by several kinases, such as protein kinase CK2 [80], activated Src kinases [81], ROCK1 (RhoA- associated kinase) [82,83], Rak tyrosine kinase [84], GSK3β (glycogen synthase kinase β) [85], and ATM (ataxia telangiectasia mutated) [86]. The phosphorylation of the PTEN C2 domain and its C-terminal regulates the PTEN function. Specifically, the PTEN phosphorylation of Ser370 and the S/T cluster (Ser380, Thr382, The383, Ser385) (also called the A4 cluster) modulates the PTEN stability and activity [21,87]. This phosphorylation of the C-terminal tail region interferes with the electrostatic binding of PTEN to the plasma membrane, controlling the membrane translocation of PTEN [88]. Deletion in the tail impairs phosphorylation and increases PTEN activity. Serine/threonine-to-alanine substitutions of PTEN block phosphorylation and alter its stability and activity. Interestingly, aspartic acid substitutions at the phosphorylation sites will restore the stability of PTEN [87].

PTEN methylation at its protein level is found in the cytoplasm and the nucleus, regulating the PTEN function. At DSB sites, NSD2 (MMSET/WHSC1) mediates the dimethylation of PTEN at K349 [89]. Methylated PTEN allows PTEN to be recruited into DNA-damage sites by the 53BP1 Tudor domain. By studying wild-type PTEN and the PTEN mutants G129E, C124S, and Y138L, it was confirmed that protein-phosphatase activity is required for efficient DSB repair, along with the dimethylation of PTEN at K349. A recent study reported that PRMT6, which is a protein arginine methyltransferase, physically associates with PTEN in the cytoplasm and methylates PTEN on R159, regulating the PI3K/AKT pathway. Moreover, this arginine dimethylation of PTEN regulates the PTEN function via pre-mRNA alternative splicing. The PTEN mutant R159K loses its lipid-phosphatase activity and is unable to dephosphorylate PIP3. PTEN-mediated tumor suppression is impaired when PTEN R159 methylation is lost [90]. 

Ubiquitination is an essential posttranslational modification for PTEN [91]. NEDD4-1 was identified as the major E3 ubiquitin ligase of PTEN; NEDD4-1 ubiquitinates PTEN and regulates tumorigenesis. As NEDD4-1 negatively regulates PTEN, the deletion of NEDD4-1 inhibits tumor growth in a PTEN-dependent manner [92]. K289 and K13 are major sites for PTEN monoubiquitination. These two lysine sites are critical for PTEN nuclear shuttling and import. The PTEN function can be suppressed by the ubiquitin ligase WWP1 (WW domain-containing ubiquitin E3 ligase via nondegradative ubiquitination [93]). 

The inactivation of PTEN by mutation and posttranslational modification, or the reduction in the PTEN protein levels by LOH and epigenetic mechanisms, results in the activation of PI3K/AKT signaling by the loss of PTEN lipid phosphatase as a major driver for susceptibility in various human cancers [94]. PTEN protein phosphatase has been proposed to dephosphorylate focal adhesion kinase (FAK) [48], c-Src [95], and PTEN itself [21], thereby inhibiting cell adhesion and migration, and it has been implicated in the maintenance of genomic integrity [14,96]. Recent studies suggest that the noncatalytic activities of PTEN with PTEN-protein-phosphatase substrates contribute to its tumor-suppressor function through poorly defined mechanisms. 

## 4. PTEN-Protein-Phosphatase Substrates and PI3K-Independent Function 

As previously indicated, PTEN functions as both a lipid phosphatase and protein phosphatase. Its lipid-phosphatase activity is well documented to convert phosphatidylinositol (3,4,5)- trisphosphate (PIP3) to phosphatidylinositol 4,5- bisphosphate (PIP2) and antagonize the PI3K/Akt pathway, thereby inhibiting cell proliferation, survival, and migration [97]. PTEN-protein-phosphatase activity is less well studied and understood. Recent studies have reported that PTEN protein phosphatase could dephosphorylate various protein substrates and play a role independent of PI3K/AKT signaling [98]. We here review the potential PTEN-protein-phosphatase substrates and their functions (see Table 2).

### 4.1. PTEN 

PTEN has been reported to play a vital role in regulating cell migration, invasion, and metastasis [48,115,116,117,118]. As indicated above, the naturally derived PTEN point mutant G129E [9,118,119], which loses lipid but maintains protein-phosphatase activity, retains the ability to inhibit cell migration and invasion, as well as metastasis [48]. To study the other mechanisms involved in cell migration, Reftopoulou and colleagues found that PTEN protein phosphatase is required for migration. They further demonstrated that PTEN protein phosphatase dephosphorylates its C domain at Thr383, inhibiting the migration independent of its effects on the PI3K pathway [21]. Interestingly, the recent studies shed light on and demonstrate that an intracellular scaffold protein (β-arrestin) and molecular assembly (ARHGAP21) interacted with dephosphorylated PTEN at Thr383, thereby negatively regulating the aggressive morphology phenotype and migration [120,121]. Another group later proposed that PTEN-protein-phosphatase activities could dephosphorylate its C domain at Thr366 and regulate cell invasion independently of PI3K/AKT signaling [49], which suggests that PTEN protein phosphatase is important for its auto-dephosphorylation, and for the autoregulation and regulation of cell migration and invasion. 

### 4.2. Abi1

Abl-interactor 1 (Abi1) is a core component of the WASP-family verprolin homologous protein (WAVE) regulatory complex (WRC). Abi1 acts as a core scaffold protein to mediate the membrane recruitment and stabilization of WRC subunits [122,123], and it is regulated by extracellular cues and intracellular signaling pathways. PTEN can bind and dephosphorylate Abi1at Y213 and S216, triggering its degradation through the calpain pathway, thereby promoting epithelial differentiation and polarization [99], as well as epithelial–mesenchymal transition and cancer-stem-cell activity [124]. PTEN dephosphorylates Abi1 and downregulates the WRC at the cell cortex, thereby reorganizing the actin cytoskeleton to facilitate the formation of the apical actin belt and adherent junctions. 

### 4.3. Β-Catenin

Transforming growth factor β (TGF-β) is a pleiotropic cytokine that plays a role in growth suppression in normal epithelial cells, but it supports metastasis formation in many tumors [125]. An early study showed that TGF-β regulates PTEN expression [6]. The study by Vogelmann and colleagues demonstrated that PTEN negatively functions downstream of the TGF-induced E-cadherin/β-catenin-mediated cell–cell-adhesion signaling pathways by reducing the TGFβ-induced phosphorylation of β-catenin [100]. However, this study showed that the epithelial dedifferentiation and dissociation of the E-cadherin adhesion complex induced by TGFβ depended on phosphatidylinositol 3-kinase (PI3-kinase) and the phosphatase PTEN. It also did not confirm which PTEN phosphatase was involved. 

### 4.4. Cofilin-1 

Cofilin-1 is an essential actin regulator. As an actin-depolymerizing factor (ADF)/cofilins family protein, Cofilin-1 regulates the rapid depolymerization of actin microfilaments that give actin its characteristic dynamic instability and its central role in muscle contraction, cell motility, and transcription regulation [126,127,128]. The activity of cofilin is regulated by a variety of mechanisms, including specific phosphorylation and dephosphorylation [128]. Serezani and colleagues reported that the dephosphorylation and activation of cofilin-1 were required for the prostaglandin E2 (PGE2)-mediated inhibition of the phagocytosis of fungi [101]. To discover the mechanism by which phosphatase proteins might be involved to dephosphorylate Coflin-1, they examined the well-known cofilin-1 phosphorylation negative regulators Lim domain kinase (LIMK) and the phosphatase slingshot-1 (SSH1) [129,130], and they found that neither of them are involved in the regulation of the dephosphorylation of cofilin-1 [101]. Engagingly, they identified that PTEN-protein-phosphatase activity directly dephosphorylates cofilin-1 at the Ser 3 site, and it mediated the PGE2-dependent activation of cofilin-1. However, this finding, and their previous results on the PTEN-lipid-phosphatase activity in the PGE2 inhibition of yeast phagocytosis [131], suggest that both PTEN lipid and protein phosphatase may involve the process of PGE2-mediated fungi phagocytosis. 

### 4.5. CREB 

The transcription factor cyclic AMP response element-binding protein (CREB) is activated via phosphorylation at serine133, which mediates gene transcription and promotes cell proliferation and survival. PTEN protein phosphatase is required for the dephosphorylation of CREB in the nucleus. Under PTEN-deficient conditions, CREB phosphorylation is enhanced independently of the PI3K/AKT pathway. The inhibition of the PI3K/AKT pathway does not affect the CREB phosphorylation in PTEN-deficient cells. C124S-mutant PTEN cannot dephosphorylate CREB, while G129E and wild-type PTEN can, which demonstrates that the protein-phosphatase activity of PTEN is essential for dephosphorylating and colocalizing with CREB in the nucleus [102], which suggests that PTEN regulates gene expression through this mechanism. 

### 4.6. Drebrin 

Drebrin is a protein that is encoded by the DBN1 gene. It is a crucial regulator of the actin cytoskeleton in neuronal cells for synaptic plasticity, neurogenesis, and neuronal migration, as well as in cancer cells for tumor invasion [132,133,134]. The defect of Drebrin in expression and activation contributes to the pathogenesis. For example, a lack of Drebrin leads to the dysfunction of the cell–cell communication, which results in the aberrant migration of metastatic cancer cells, the aberrant synaptic function in dementia, and the rupture of the endothelial integrity and memory disturbance in Alzheimer’s disease [133,135]. The increase in Drebrin at Ser 647 is linked to cognitive deficits and seizures [103]. To study the mechanism of Drebrin on the regulation of the central synapse, Kreis et al. [103] demonstrated that PTEN could bind to Drebrin, and this results in its dephosphorylation at the Ser 647 site. However, the dephosphorylation of Drebrin was only confirmed by using a PTEN phosphatase dead mutant form of PTEN C124S, which cannot rule out its lipid-phosphatase activity. 

### 4.7. Dvl 

PTEN is an essential regulator of multicilia formation and cilia disassembly via Dishevelled (DVL2) phosphorylation. DVL is an important component of the WNT signaling pathways that plays a role during convergent extension movements. DVL is a ciliogenesis regulator in *Xenopus* and human epithelial cells, and it has been identified as a direct substrate for PTEN. Among the DVL proteins, DVL2 and DVL3 have the strongest associations with PTEN. The knockdown of PTEN increases the DVL2 phosphorylation on serine143 during cilia formation. By studying wild-type PTEN and mutants of PTEN (C124S, G129E, Y138L), it was confirmed that the protein-phosphatase activity of PTEN is responsible for directly dephosphorylating DVL2 on serine 143 [104], which implicates PTEN in multicilia formation and movement.

### 4.8. FAK 

Focal adhesion kinase (FAK) is one of the earliest confirmed substrates for PTEN. PTEN overexpression inhibits cell migration via reducing the phosphorylation of FAK. The effect of PTEN on cell spreading was examined on fibronectin (FN) in multiple cell lines: NIH 3T3 cells, human fibroblast cells, DBTRG-05MG cells (glioblastoma), and U-87MG (glioma) cells. The overexpression of PTEN delayed or inhibited the spreading in these cell lines [48]. FAK was first found to be a substrate of the Src proto-oncogene. It is an important regulator of cell adhesion and motility. The tyrosine phosphorylation of FAK is associated with focal contacts that form at ECM integrin junctions, and integrin-binding proteins recruit FAK to the focal contacts. It has been reported that PTEN suppresses cell migration, invasion, and metastasis through the dephosphorylation of FAK at Tyr397 [48,105]; however, specific FAK dephosphorylation sites have not been observed in other cell types [136], and other mechanistic details have yet to be uncovered. 

### 4.9. Glucocorticoid Receptor (GR) 

GR is a pleiotropic nuclear receptor and transcriptional regulatory factor that controls the network of glucocorticoid (GC)-responsive genes in a positive or negative manner for regulating numerous physiological and cellular processes [137]. In cancer, GR activation also appeared as tumor-suppressing [138] and tumor-promoting effects [139]. A recent study in breast cancer development discovered the link between the GR function and PI3K-pathway activation, which explains the contrasting results of GR in breast cancer [52]. Using knock-in (KI) mice harboring wild-type PTEN, a PTEN lipid phosphatase-deficient mutant (PTEN G129E), or a PTEN phosphatase dead mutant (PTEN C124S), Yip et al. found that, although the loss of the PTEN-lipid-phosphatase function cooperates with oncogenic PI3K to promote rapid mammary tumorigenesis, the additional loss of PTEN-protein-phosphatase activity triggered an extensive cell-death response that was evident in early and advanced mammary tumors, which points out that the dual regulation of GR by PI3K and PTEN functions as a rheostat. Notably, they confirmed that wild-type PTEN and PTEN G129E expression could not alter the phosphorylation of GR at Ser211, while the PTEN C124S upregulates the phosphorylation of GR at Ser211 in mammary epithelial cells, which suggests that PTEN-protein-phosphatase activity is required for the dephosphorylation of GR at Ser211, and that the GR dephosphorylation status acts a switch that is mediated by PTEN phosphatase activity in PI3K-induced tumorigenesis. 

### 4.10. IRS1 

PTEN protein phosphatase can dephosphorylate insulin receptor substrate-1 (IRS1). IRS1 is a mediator of insulin and IGF signaling, which is negatively affected by NEDD4. In NEDD4-deficient cells, IGF signaling becomes defective, while AKT activation is unimpaired. PTEN ablation rescues impaired IGF signaling by dephosphorylating IRS1. Although NEDD4 is required for IGF signaling, the role of NEDD4 in IGF signaling is PTEN-dependent. NEDD4 inhibits the PTEN function to enable IGF signaling. NEDD4 is responsible for the ubiquitination of PTEN, and for suppressing its phosphatase activity. The C124S mutant of PTEN is not able to dephosphorylate IRS1, while the G129E mutant and wild-type PTEN are, which proves that IRS1 is a direct substrate of PTEN’s protein phosphatase [106]. 

### 4.11. MCM2

The MCM2-7 complex is one of the core components of the replisome [140,141], and it plays crucial roles in replication origin firing, elongation, termination, and the replication-stress response [142,143]. MCM2 is a critical component of the MCM2-7 complex, and it is a core replication helicase of the replisome that is regulated by the MCM2 phosphorylation status. A recent study showed that PTEN physically interacts with MCM2 and is involved in DNA replication regulation. In response to replication stress, PTEN dephosphorylates MCM2 at serine41 through its protein phosphatase, which limits DNA replication for progression and prevents chromosomal aberration. This suggests that PTEN is essential for maintaining genomic stability, and that the loss of PTEN affects the DNA-replication defects and genomic instability [107]. 

### 4.12. NKX3.1

NKX3.1, which is a prostate-specific homeobox gene, is a gatekeeper suppressor and is commonly deleted in prostate cancer. NKX3.1 inhibits cell proliferation and mediates cell apoptosis and DNA repair. Bowen et al. discovered that PTEN is an NKX3.1 phosphatase substrate that protects NKX3.1 by opposing phosphorylation at serine185. Phosphorylation at NKX3.1 S185 causes polyubiquitination and proteasomal degradation, and so PTEN extends the NKX3.1 half-life and prevents NKX3.1 from degradation. Specifically, PTEN localizes in the nucleus, interacts with NKX3.1, and directly prevents the antagonistic kinase DYRK1B from phosphorylating NKX3.1. NKX3.1 is a protein-phosphatase substrate of PTEN because G129E-mutant PTEN affected the NKX3.1 half-life while C124S and Y128L did not [108]. 

### 4.13. PLK1 

Polo-like kinase 1 (PLK1) is a mitotic kinase that regulates mitotic entry and exit. PLK1 controls spindle bipolarity and is involved in cytokinesis. PTEN physically associates with PLK1 and dephosphorylates PLK1, maintaining genomic stability during cell division. PTEN loss or deficiency causes failure in cytokinesis through nondisjunction chromosomes and cleavage-furrow regression, which leads to spontaneous polyploidy and resistance to spindle disruption. In addition, PTEN loss results in PLK1 hyperphosphorylation on residue Thr210. Therefore, PTEN inhibits PLK1 through its protein phosphatase, and the PTEN phosphatase function is essential for restricting polyploidization and restoring sensitivity to spindle drugs [109]. 

### 4.14. PTK6 

PTEN inhibits protein tyrosine kinase 6 (PTK6/BRK/Sik) activity in prostate cancer cells by dephosphorylating PTK6 at tyrosine 342 (PY342). In the absence of PTEN, PTK6 is activated and downstream oncogenic signaling is promoted [111]. A recent study demonstrates that PTK6 PY342 is a PTEN-protein-phosphatase substrate. Wild-type PTEN and a lipid-phosphatase-effective PTEN mutant retained the protein-phosphatase-activity target phosphorylation of PTK6 at Y342. In contrast, the PTEN Y138L mutant that lacks protein-phosphatase activity and two different catalytically inactive PTEN mutants did not target PTK6 Y342. 

### 4.15. Pol II

The RNA polymerase II (Pol II) C-terminal domain (CTD) constantly undergoes cycles of phosphorylation/dephosphorylation during gene transcription. It was demonstrated that PTEN dephosphorylates Pol II CTD with specificity for Ser5, and that the phosphorylation of Pol II CTD Ser5 is inversely related to PTEN expression. When there is an overexpression of PTEN, there will be a decrease in the Pol II CTD phosphorylation; global Pol II CTD phosphorylation increases along with the PTEN loss. Pol II CTD is a significant platform for posttranslational modifications, such as elongation, termination, and co-transcriptional processes. PTEN, as a Pol II CTD phosphatase, raises the possibility that PTEN can regulate global transcription [110]. However, this study did not confirm that the dephosphorylation of Pol II CTD at Ser5 is through the PTEN protein phosphatase.

### 4.16. Rab7

Rab7 is a GTPase for endosome maturation that is involved in epidermal growth factor receptor (EGFR) signaling. PTEN dephosphorylates Rab7 on two residues, S72 and Y183, and it promotes late endosome maturation, which reduces EGFR signaling. The residues are required to associate Rab7 with GDP dissociation inhibitor (GDI)-dependent recruitment to late endosomes and maturation. EGFR is a receptor tyrosine kinase that regulates cell proliferation, growth, and motility. PTEN controls the EGFR-endocytic-trafficking pathway via the dephosphorylation of Rab7 and the localization of Rab7. The loss of PTEN causes EGFR transport from early to late endosomes because PTEN is needed for Rab7-endosomal-membrane targeting. PTEN-mediated Rab7 dephosphorylation allows Rab7 to interact with GDI, GEF, and effector proteins. Rab7 has been identified as a protein substrate for PTEN, which provides a new mechanism for controlling the EGFR signaling in cells via PTEN [112]. 

### 4.17. Shc

Src homology collagen (Shc) is a direct substrate for PTEN protein phosphatase. Shc is an SH2-binding adaptor molecule that activates the Raf and MAPK pathway. PTEN inhibits the MAPK pathway by dephosphorylating Shc in positions Tyr239/240, which reduces cell proliferation and inhibits Shc-mediated tumor metastasis in renal-cell carcinoma (RCC) [113]. These results show a critical role for PTEN in the metastasis of RCC that is dependent on the protein-phosphatase activity via Shc. This new insight opens an array of different approaches that complement the current cancer therapy and metastasis prediction in RCC. 

### 4.18. SRC

SRC is a membrane-anchored tyrosine kinase that is activated following the engagement of many different classes of cellular receptors, and it regulates various biological activities, including cell proliferation, adhesion, migration, and transformation [144]. The study reported that SRC modulates the antibody Trastuzumab that targets the human epithermal growth factor receptor-2 (HER-2 or ERBB2) response in breast cancer, and that activating SRC by phosphorylation at Tyr416 was required for regulating the multiple resistance pathways [114]. Interestingly, they later found that PTEN-protein-phosphatase activity could dephosphorylate the SRC at Tyr416, but its lipid-phosphatase activity did not, which suggests that the SRC Tyr 416 is a direct substrate for PTEN-protein-phosphatase activity, and that PTEN-protein-phosphatase activity negatively regulates the SRC-mediated drug-resistant signaling pathway. 

## 5. Summary and Perspective

Since the discovery of PTEN in 1997, the knowledge of its biology and function has been continuously growing. The negative regulation of the PI3K/AKT signaling pathway by PTEN lipid phosphatase has been well studied as the main driver for tumorigenesis and the physiologic function of PTEN. Recently identified protein substrates of the PTEN protein phosphatase will not only explain the various specific roles of PTEN, but they will also help us to understand its biological function independent of the PI3K/AKT signaling pathway. Based on current studies, we can clearly see that the biological roles of PTEN-protein-phosphatase activity act to: (1) maintain its ‘active’ configuration and stability by dephosphorylating its C domain; (2) regulate the cell movement, migration, and invasion by the dephosphorylation of its substrates, such as Abi1, Dvl, FAK, Shc, SRC, Drebrin, cofilin-1, and β-catenin; (3) participate in the regulation of gene expression through dephosphorylating CREB, IRS1, NKX3.1, Rab7, PTK6, and Pol II; (4) maintain genomic stability by dephosphorylating MCM2 and PLK1. Moreover, a recent study showed that PTEN-protein-phosphatase activity is required to dephosphorylate GR at Ser211, and that the GR dephosphorylation status acts as a switch that is mediated by the PTEN phosphatase activity in PI3K-induced tumorigenesis [52]. However, the molecular mechanisms of PTEN protein phosphatase in the biological function are very complex. We still need more knowledge of the PTEN-protein-phosphatase substrates and their signaling networks to elucidate the molecular signaling pathways at the physiologic and pathogenic levels. Although the approaches of the PTEN knock-out and knock-in of PTEN lipid phosphatase and phosphatase mutants have determined the many substrates and potential role of PTEN protein phosphatase, new models with the knock-in of PTEN wild-type, lipid-, protein-, or both lipid- and protein-phosphatase mutant forms with similar protein levels and the same backgrounds, will more precisely facilitate the discovery of more substrates of PTEN phosphatase and their functions for figuring out this complex puzzle. Moreover, PTEN has a PDZ/BD domain that can bind with other proteins, and PTEN-binding proteins will likely also play important roles in the PTEN function beyond antagonizing PI3K/AKT signaling [145]. To date, the study of PTEN has provided a full appreciation of its complex and critical physiological and pathological functions, which indicates that future studies will prove exceedingly rewarding.

## Figures and Tables

**Figure 1 cancers-14-03666-f001:**
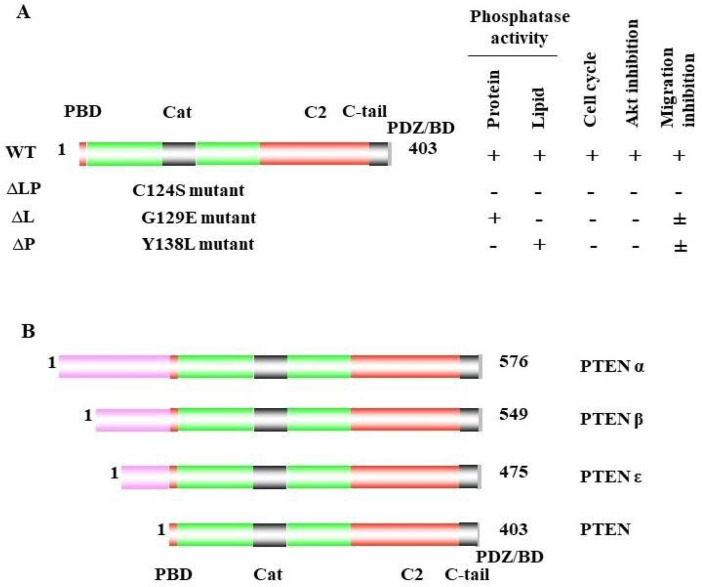
Schematic of the PTEN protein. (**A**) PTEN contains five functional domains: two key domains that are required for its tumor-suppressor function: the phosphatase (catalytic) domain (amino acids 14–189), with an active site included within the residues 123 and 130 (black), and the C2 (lipid-membrane-binding) domain (amino acids 190–350) (red); two binding domains that are the N-terminal PIP2-binding domain PBD (amino acids 1–14) and C-terminal PDZ-binding domain (grey; amino acids 401–403), which binds proteins containing PDZ domains; the carboxy-terminal region (amino acids 351–400), which contains PEST sequences and contributes to PTEN stability and activity, and is less well defined in the tumor-suppressor functions of PTEN. Wild-type PTEN with both lipid- and protein-phosphatase activity inhibits the cell cycle, AKT activity, and cell migration. The mutation at C124S (∆LP) inactivates both PTEN lipid and protein phosphatase, which provokes the loss of the inhibition of cell-cycle arrest and AKT and cell migration. The G129E (∆L) mutant loses only its lipid-phosphatase activity and can still inhibit cell migration. The mutation of Y138L is deficient in its protein phosphatase, which may lose the capacity to inhibit cell migration. (**B**) Three PTEN alternative translational isoforms, PTENα, PTENβ, and PTENε, which are produced from the same mRNA as canonical PTEN and are generated due to non-AUG translational initiation. Each has a longer N-terminal extension than the canonical PTEN protein.

**Figure 2 cancers-14-03666-f002:**
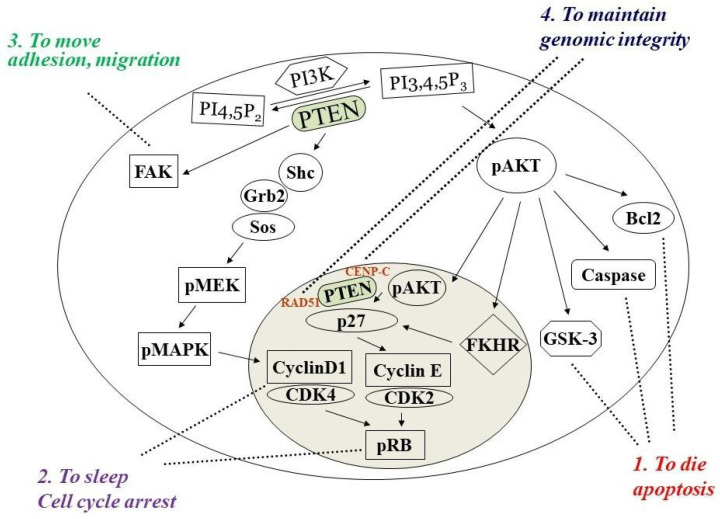
PTEN has four major cellular functions. By dephosphorylating PIP3 and negatively regulating the activation of AKT, PTEN can both prevent the activation of Bcl2 and GSK-3 and promote the activation of caspase, thereby inducing cell apoptosis (1), and against both the AKT and MAPK signaling pathways to promote cell-cycle arrest (2). The protein phosphatase dephosphorylates the FAK proteins to reduce cell adhesion, movement, and migration (3). It can also shuttle into the nucleus to maintain genomic integrity (4).

**Figure 3 cancers-14-03666-f003:**
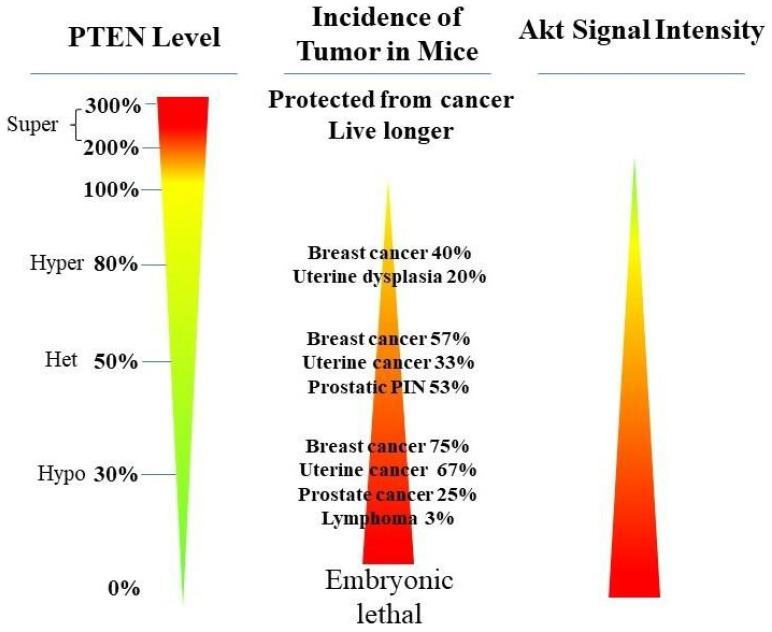
PTEN functions as a haploinsufficient tumor suppressor. PTEN loss correlates with the incidence of tumorigenesis and increased activity of AKT in a dose-dependent manner (the more loss of PTEN (**left panel**), the higher tumor incidence (**middle panel**) and AKT activity (**right panel**)). PTEN deficiency (0% of PTEN) leads to embryonic lethality and hyperactivated AKT, while the hypomorphic allelic loss of PTEN (Hypo 30%) revealed more increased cancer phenotypes than the heterozygous loss of a PTEN allele (Het 50%) and the hypermorphic allelic loss of PTEN (Hyper 80%, with small 20% reductions in PTEN doses) (the tumor incidence from high to low: Hypo 30% > Het 50% > Hyper 80%). In contrast, elevated PTEN (>100%) protects against tumorigenesis and promotes longevity in GEM models [23,24,25].

**Figure 4 cancers-14-03666-f004:**
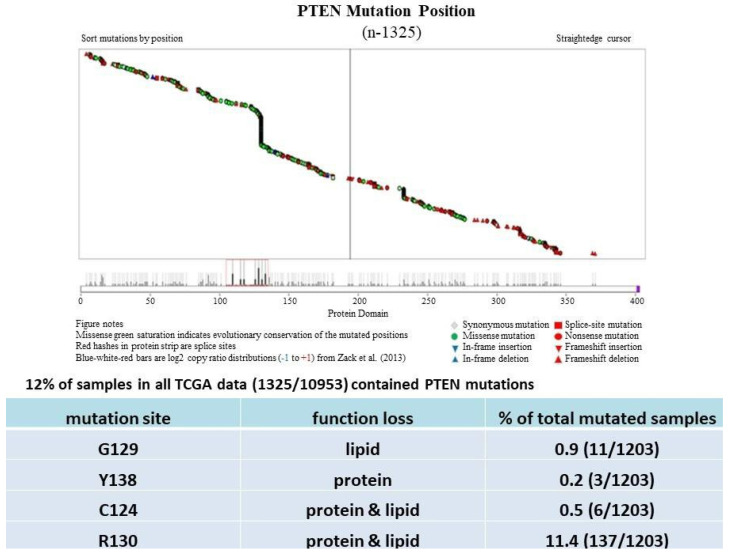
PTEN mutations cause PTEN inactivation and loss of the tumor-suppressive function. Most PTEN mutations in cancer (TCGA dataset) are found in its phosphatase domain, including G129, Y138, C124, and R130, which lose lipid or protein or both phosphatase activities.

**Figure 5 cancers-14-03666-f005:**
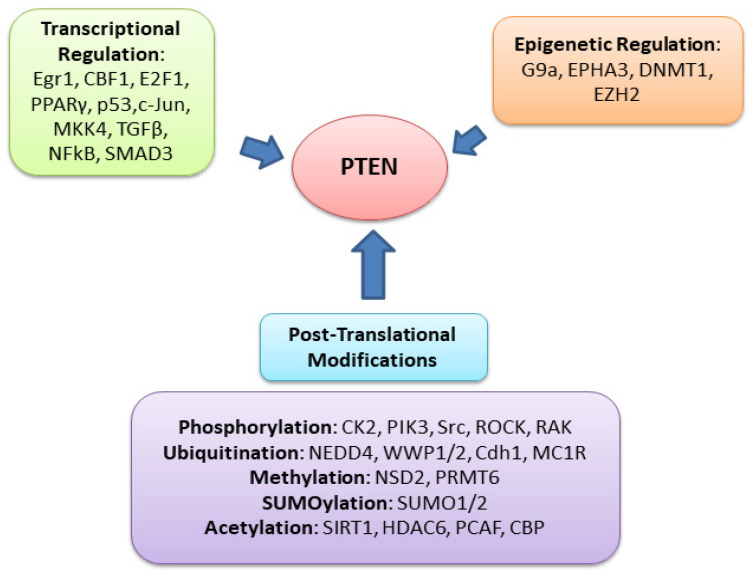
The function of PTEN can be regulated by different mechanisms through various genes, including transcriptional regulation, post-transcriptional regulation, epigenetic regulation, and posttranslational modifications. The transcriptional downregulation of PTEN causes protein levels to decrease by epigenetic regulation (promoter methylation) transcriptional factors.

**Table 2 cancers-14-03666-t002:** The putative substrates of the PTEN protein phosphatase.

Putative Substrate	Function	Cell Lines	Reference(s)
Abi1 (Y213, S126)	Mediation of membrane recruitment and stabilization of WRC subunits	U87MG, LN229	[99]
β-catenin	Regulates cell-cell adhesion	PANC-1	[100]
Cofilin-1 (Ser3)	Mediates prostaglandin E2 (PGE2) - inhibited the phagocytosis of fungi	Alveolar macrophage	[101]
CREB (S133)	Transcription factor; promotes cell survival	MDA-MB-175/468/231/415	[102]
Drebrin (Ser647)	Negatively regulates the highly synchronized neuronal network activity	HEK293T	[103]
Dvl (S143)	Stabilizes primary cilia as a regulator; WNT signaling	hTERT RPE-1 cells	[104]
FAK (Y397)	Inhibits cell spreading and migration	U-87MG, NIH/3T3	[48,105]
GR (Ser211)	Suppresses survival of PTEN and PI3K mutant cells	MEF cells	[52]
IRS1 (Y612)	Activates PIP3 production via binding to PI3K	HEK293T	[106]
MCM2 (Ser41)	Modulates fork progression in DNA replication	U2OS cells	[107]
NKX3.1 (S185)	growth suppressor; mediator of apoptosis; enhances DNA repair	Prostate cancer	[108]
PLK1 (Thr210)	Controls spindle bipolarity, mitotic entry/exit, cytokinesis	HeLa cells	[109]
Pol II (Ser5)	Global gene expression regulation	Breast-cancer cell MCF-7	[110]
PTEN (Thr383 or Thr366)	Inhibits cell migration and invasion	U87, U373	[22,49]
PTK6 (Y342)	Promotes invasive prostate cancer	PC3, DU145	[111]
Rab7 (S72)	GTPase for endosome maturation; EGFR signaling	HeLa; MDA-MB-468; HEK293T	[112]
Shc (Tyr239/240)	Adaptor molecule; activates Raf and MAPK pathway	786-O ccRCC cells	[113]
SRC (Y416)	No-receptor tyrosine kinase, regulates drug resistance	MDA-MB-468, BT474	[114]

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
