# Peer review of "PTEN Dual Lipid- and Protein-Phosphatase Function in Tumor Progression"

_cancers, 2022, doi:10.3390/cancers14153666_

Round 1

Reviewer 1 Report

In this review study, Liu and colleagues describe how the PTEN protein phosphatase contributes to its tumor-suppressive actions through PI3K-independent activities.

The topic is original and addresses a specific gap in the field. I also found this topic clinically relevant.

The article is well written. I found the conclusion to be in line with the evidence and arguments presented.

Major point:

However, the authors should update the references list with the latest research (between 2018-2022) and explain/discuss. They have cited very few studies.

Minor points:

The tables are ok but the figures.

For example, Figure 2 is not clear. The authors mentioned in the text (page 3, line 76) "decreasing the level of cyclin D1" but in the figure, there is Cyclin E/cdk2.

The caption of figure 3 is not clear.

Figure 4 is very poor quality.

Author Response

Dear Editors and Reviewers,

Thank you for your correspondence of 20th June 2022, communicating the comments from three expert reviewers. We much appreciate the receiving Editor and reviewers’ comments and suggestions for improving our manuscript. We are also pleased with the positive reviews. We have carefully considered these critiques and made significant changes to our current manuscript. We hope that the revised manuscript is now suitable for publication in Cancers.

Specific responses to the points raised by the reviewers are as follows: Our responses are in the bold courier font for easier visualization.

Reviewer #1 (Comments and Suggestions for Authors):

In this review study, Liu and colleagues describe how the PTEN protein phosphatase contributes to its tumor-suppressive actions through PI3K-independent activities.

  • The topic is original and addresses a specific gap in the

field. I also found this topic clinically relevant.

  • The article is well written. I found the conclusion to be in line with the evidence and arguments

We thank the reviewer for these positive comments.

Major point:

However, the authors should update the references list with the latest research (between 2018-2022) and explain/discuss. They have cited very few studies.

We thank the reviewer for these great suggestions. We now updated the references list and discussion.

Minor points:

The tables are ok but the figures.

37 Convent Drive, Room 5046, Bethesda, MD 20892-4264 |[email protected] | 240 760 6812,

301 480-4662 (FAX)

For example, Figure 2 is not clear. The authors mentioned in the text (page 3, line 76) "decreasing the level of cyclin D1" but in the figure, there is Cyclin E/cdk2.

The caption of figure 3 is not clear. Figure 4 is very poor quality.

We appreciate this reviewer’s suggestion for improving our manuscript. We updated the figures.

Reviewer 2 Report

This review from Liu et al aims to provide an overview on the contribution of PTEN protein phosphatase activity to tumorigenesis.

The review lists a number of reported phospho-protein substrates which, acting downstream of PTEN in different ways, suppress tumour progression and control cell migration in a PIP3-independent manner. The overall scope of the manuscript is interesting, and the list of PTEN protein-targets quite comprehensive; however, despite the long list of reported proteins (13 included), it is surprising to read in the final “Summary and perspective” that “PTEN protein phosphatase substrate signalling network is still limited” and that “new knock-in models are needed to discover new substrates”.

This may still be the case but nevertheless, the review does not address what is the general view on PTEN protein phosphatase activity versus the lipid function. Authors should provide a conclusive synthesis and clarify what is the overall consensus on the biological role of PTEN protein phosphatase activity. This is not clear from the current status of the manuscript, despite stating in the abstract that this was its purpose. Moreover, functionally relevant knock-in mice for PTEN have been generated (e.g. Wang H, PNAS 2010; Papa A et al, Cell 2014) and conclusions from those studies with respect to the role of PTEN protein phosphatase activity should be discussed.

The review would also benefit from a major revision on multiple additional aspects which are detailed below:

1- The manuscript is organized in multiple paragraphs which at times contain repetitive sentences that could be eliminated for simplicity, and incorrect statements which should be corrected: see page 2, row 69, “PTEN localizes in the cytoplasm to perform its lipid phosphatase activity”, maybe the authors meant cytoplasmic membrane?

Also, is the C-terminal PDZ domain of PTEN the only one interacting with other proteins (page 1, row 33)?

Please read and amend all.

2- Please review the reference list and include missing references at the end of each sentence. For example, page 7, row 204, what is the reference to papers showing that PTENG129E still inhibits cell spreading and mobility? On the next paragraph, what is the reference to papers showing that the PTENG130 causes loss of PTEN lipid and protein phosphatase activity? Please read the entire manuscript and fix all missing references.

Some references have duplications, e.g. ref 22 and 45.

3- Some reported PTEN protein phosphatase substrates have been missed:

i) Zhang S et al, 2011 Nature Medicine, “Combating trastuzumab resistance by targeting SRC, a common node downstream of multiple resistance pathways”

ii) Zhang F, Cancer Discovery 2019, “PTEN Methylation by NSD2 Controls Cellular Sensitivity to DNA Damage”

iii) Yip HYK et al, Mol Cell 2020 “Control of Glucocorticoid Receptor Levels by PTEN Establishes a Failsafe Mechanism for Tumour Suppression”

4- Figure 4 should be provided in higher resolution as text is almost illegible.

Author Response

Dear Editors and Reviewers,

Thank you for your correspondence of 20th June 2022, communicating the comments from three expert reviewers. We much appreciate the receiving Editor and reviewers’ comments and suggestions for improving our manuscript. We are also pleased with the positive reviews. We have carefully considered these critiques and made significant changes to our current manuscript. We hope that the revised manuscript is now suitable for publication in Cancers.

Specific responses to the points raised by the reviewers are as follows: Our responses are in the bold courier font for easier visualization.

This review from Liu et al aims to provide an overview on the contribution of PTEN protein phosphatase activity to tumorigenesis.

The review lists a number of reported phospho-protein substrates which, acting downstream of PTEN in different ways, suppress tumour progression and control cell migration in a PIP3- independent manner. The overall scope of the manuscript is interesting, and the list of PTEN protein-targets quite comprehensive; however, despite the long list of reported proteins (13 included), it is surprising to read in the final “Summary and perspective” that “PTEN protein phosphatase substrate signalling network is still limited” and that “new knock-in models are needed to discover new substrates”.

We appreciate the reviewer for these positive comments and updated our statements.

This may still be the case but nevertheless, the review does not address what is the general view on PTEN protein phosphatase activity versus the lipid function. Authors should provide a conclusive synthesis and clarify what is the overall consensus on the biological role of PTEN protein phosphatase activity.

This is not clear from the current status of the manuscript, despite stating in the abstract that this was its purpose.

Moreover, functionally relevant knock-in mice for PTEN have been generated (e.g. Wang H, PNAS 2010; Papa A et al, Cell 2014) and conclusions from those studies with respect to the role of PTEN protein phosphatase activity should be discussed.

These are great suggestions. Based on current studies, we provided the consensus conclusions for the biological function of PTEN protein phosphatase activity (see section 5) and added more discussion of the studies on PTEN mutant knock-in mice in the revised manuscript.

The review would also benefit from a major revision on multiple additional aspects which are detailed below:

  • The manuscript is organized in multiple paragraphs which at times contain repetitive sentences that could be eliminated for simplicity, and incorrect statements which should be corrected: see page 2, row 69, “PTEN localizes in the cytoplasm to perform its lipid phosphatase activity”, maybe the authors meant cytoplasmic membrane?

Also, is the C-terminal PDZ domain of PTEN the only one interacting with other proteins (page 1, row 33)?

Please read and amend all.

We thank the reviewer for pointing out these errors. We have corrected them in our revised manuscript.

  • Please review the reference list and include missing references at the end of each sentence. For example, page 7, row 204, what is the reference to papers showing that PTENG129E still inhibits cell spreading and mobility? On the next paragraph, what is the reference to papers showing that the PTENG130 causes loss of PTEN lipid and protein phosphatase activity? Please read the entire manuscript and fix all missing references.

Some references have duplications, e.g. ref 22 and 45.

We thank the reviewer for pointing out the important issue. We have added the reference and corrected the reference list.

  • Some reported PTEN protein phosphatase substrates have been missed:
  1. Zhang S et al, 2011 Nature Medicine, “Combating trastuzumab resistance by targeting SRC, a common node downstream of multiple resistance pathways”
  2. Zhang F, Cancer Discovery 2019, “PTEN Methylation by NSD2 Controls Cellular Sensitivity to DNA Damage”
  • Yip HYK et al, Mol Cell 2020 “Control of Glucocorticoid Receptor Levels by PTEN Establishes a Failsafe Mechanism for Tumour Suppression”

We thank the reviewer for providing the important information for improving our manuscript. We have added SRC and GR as the important substrates of PTEN protein phosphatase in Table 2 and revised the manuscript. However, Zhang’s study (Cancer Discovery,2019) showed that NSD2 was a PTEN regulator to mediate the methylation of PTEN at K349. We cited and discussed this paper in section 3 of our manuscript.

  • Figure 4 should be provided in higher resolution as text is almost

We have updated this figure in our revised manuscript.

Reviewer 3 Report

The review by Liu et al. provides a general introduction on PTEN function, modes of regulation, covering alternative variants and the different mutants used to pick apart lipid/protein phophatase activities. This is followed by a focus on the less well-known functional effects of PTEN protein phosphatase activity. In general I think it's well constructed and covers most of the important points. However, more effort should be put into improving the quality of some of the figures and legends, as well as some parts of the text, so it is clearer and doesn't lead to confusion. This is detailed below.

Line 13 - add "activity" after phosphatase

Line 13 - add "activity" after phosphatase

Line 33 - once the PDZ-binding domain is defined as PDZ/BD it should be used thereafter. For example, in the same sentence it is then referred to as a PDZ domain, which is wrong. Use the PDZ/BD abbreviation here and also elsewhere in the text. Replace "could" with "can." Also, add recent reviews covering protein partners that bind the PDZ/BD, e.g. PMID: 31932468 PMID: 25843297

Line 38 - take out "the" and add "substrates" after "protein" then take out "as a"

Figure 1 - Is it not Y138L mutant instead of Y138E???? In the figure legend line 61 take out "which loss the inhibition" and replace with "which provokes the loss of the inhibition". Line 62- replace all of this sentence with "The G129E(∆L) mutant loses only its lipid phosphatase activity and can still inhibit cell migration". Line 63 - Replace "Mutation of Y138E" with "The Y138L mutant" and then replace "which may lose the inhibition of cell migration" with "which may lose the capacity to inhibit cell migration" Line 64 expand the figure legend (B) here to explain/provide detail on the pink regions on the alternative isoforms.

Line 96-97. change "by dephosphorylating threonine 383" to "following autodephosphorylation of threonine 383"

Line 113 replace "lethal" with "lethality"

Line 129 - Ref (17) is wrong please replace with correct ref.

The quality of Figure 4 is not acceptable, it is completely blurred and needs improved.

Table 1 - I'm not sure these references are in the reference list at the end. Also they are not in the numbered reference format anyway. Please rectify all of this.

Lines 229-238. All of the papers aren't referenced, either they should be put in or reviews cited e.g. PMID: 21236500. Also, there's not much relation to this and Figure 5 - please make it fit. MC1R is a GPCR and not a transcription factor...PTEN has an effect on MTF1 and AIB1 activities and not the other way around. Are the epigenetic regulations in the figure discussed somewhere? Should the figure not have a post-transcriptional regulation panel in relation with the text 229-238 mentioning miRs... In conclusion, I think things are best kept simple here and the figure should reflect factors that influence PTEN function (and not the other way around) and it should reflect the text - p53, EGR-1, PPAR- miRs etc. Also, the PDZ-BD of PTEN is acetylated but it's not an enzyme as suggested in the purple box. This figure and legend should be substantially improved.

-Line 247 replace "A4" with "S/T

-Line 273 replace "the" with "a"

-Line 276 replace "ubiquitination" with "monoubiquitination"

-Line 277 - add "nuclear" before "shuttling"

-Line 278 - add "via non-degradative ubiquitination" at the end of the sentence.

Line 288 "PTEN protein phosphatase substrates and PI3K independent functions in metastasis." Not all of the substrates mentioned are implicated in cell migration/metastasis. Either change the title and take out "in metastasis" or regroup the substrates implicated in migraton/invasion/metastasis and discuss the others in another section.

In the PTEN section, dephosphorylation of PTEN at Thr383 has been shown to control its association with the molecular scaffolds beta-arrestins with conseqeunces for glioma cell migration PMID: 21642958 and multicellular assembly PMID: 28749339; this could be discussed and added here. Also, some protein substrates haven't been discussed and should be added - these include Drebrin (Kreis et al. 2013 PLoS ONE), Beta-catenin (Vogelmann et al. 2005 J Cell Sci) and cofilin (Serezani et al 2012 Sci Signaling).

PTEN should also be added to the table 2. Cell lines used were U87 and U373. The references here are also not in "numbered" format.

Line 304 - replace "remains" with "retains"

Line 450 - replace BDZ with PDZ-BD

Author Response

Dear Editors and Reviewers,

Thank you for your correspondence of 20th June 2022, communicating the comments from three expert reviewers. We much appreciate the receiving Editor and reviewers’ comments and suggestions for improving our manuscript. We are also pleased with the positive reviews. We have carefully considered these critiques and made significant changes to our current manuscript. We hope that the revised manuscript is now suitable for publication in Cancers.

Specific responses to the points raised by the reviewers are as follows: Our responses are in the bold courier font for easier visualization.

The review by Liu et al. provides a general introduction on PTEN function, modes of regulation, covering alternative variants and the different mutants used to pick apart lipid/protein phophatase activities. This is followed by a focus on the less well-known functional effects of PTEN protein phosphatase activity. In general I think it's well constructed and covers most of the important points.

We appreciate the reviewer for these positive comments.

However, more effort should be put into improving the quality of some of the figures and legends, as well as some parts of the text, so it is clearer and doesn't lead to confusion. This is detailed below.

Line 13 - add "activity" after phosphatase Line 13 - add "activity" after phosphatase

Line 33 - once the PDZ-binding domain is defined as PDZ/BD it should be used thereafter. For example, in the same sentence it is then referred to as a PDZ domain, which is wrong. Use the PDZ/BD abbreviation here and also elsewhere in the text. Replace "could" with "can." Also, add recent reviews covering protein partners that bind the PDZ/BD, e.g. PMID: 31932468 PMID: 25843297

Line 38 - take out "the" and add "substrates" after "protein" then take out "as a"

Figure 1 - Is it not Y138L mutant instead of Y138E???? In the figure legend line 61 take out "which loss the inhibition" and replace with "which provokes the loss of the inhibition". Line 62- replace all of this sentence with "The G129E(∆L) mutant loses only its lipid phosphatase activity and can still inhibit cell migration". Line 63 - Replace "Mutation of Y138E" with "The Y138L mutant" and then replace "which may lose the inhibition of cell migration" with "which may lose the capacity to inhibit cell migration" Line 64 expand the figure legend (B) here to explain/provide detail on the pink regions on the alternative isoforms.

Line 96-97. change "by dephosphorylating threonine 383" to "following autodephosphorylation of threonine 383"

Line 113 replace "lethal" with "lethality"

Line 129 - Ref (17) is wrong please replace with correct ref. The quality of Figure 4 is not acceptable, it is completely blurred and needs improved.

Table 1 - I'm not sure these references are in the reference list at the end. Also they are not in the numbered reference format anyway. Please rectify all of this.

Lines 229-238. All of the papers aren't referenced, either they should be put in or reviews cited e.g. PMID: 21236500. Also, there's not much relation to this and Figure 5 - please make it fit. MC1R is a GPCR and not a transcription factor...PTEN has an effect on MTF1 and AIB1 activities and not the other way around. Are the epigenetic regulations in the figure discussed somewhere? Should the figure not have a post-transcriptional regulation panel in relation with the text 229-238 mentioning miRs... In conclusion, I think things are best kept simple here and the figure should reflect factors that influence PTEN function (and not the other way around) and it should reflect the text - p53, EGR-1, PPAR- miRs etc. Also, the PDZ-BD of PTEN is acetylated but it's not an enzyme as suggested in the purple box. This figure and legend should be substantially improved.

-Line 247 replace "A4" with "S/T

-Line 273 replace "the" with "a"

-Line 276 replace "ubiquitination" with "monoubiquitination"

-Line 277 - add "nuclear" before "shuttling"

-Line 278 - add "via non-degradative ubiquitination" at the end of the sentence.

Line 288 "PTEN protein phosphatase substrates and PI3K independent functions in metastasis." Not all of the substrates mentioned are implicated in cell migration/metastasis. Either change the title and take out "in metastasis" or regroup the substrates implicated in migraton/invasion/metastasis and discuss the others in another section.

In the PTEN section, dephosphorylation of PTEN at Thr383 has been shown to control its association with the molecular scaffolds beta-arrestins with conseqeunces for glioma cell migration PMID: 21642958 and multicellular assembly PMID: 28749339; this could be discussed and added here. Also, some protein substrates haven't been discussed and should be added - these include Drebrin (Kreis et al. 2013 PLoS ONE), Beta-catenin (Vogelmann et al. 2005 J Cell Sci) and cofilin (Serezani et al 2012 Sci Signaling).

PTEN should also be added to the table 2. Cell lines used were U87 and U373. The references here are also not in "numbered" format.

Line 304 - replace "remains" with "retains" Line 450 - replace BDZ with PDZ-BD

We thank the reviewer very much for pointing out all the important issues. We updated these points in our revised manuscript.

We hope our revised manuscript is satisfactory to all reviewers and is now acceptable for publication in Cancers.  Thank you for considering our work for your journal.

Round 2

Reviewer 2 Report

I thank the authors for updating their review. Some of the comments raised during the initial revision have been addressed, but there are still critical issues that make the review difficult to understand.

The accuracy of some paragraphs (please see row 226 to 244, this paragraph is disconnected from the following paragraph even though they discuss the same set of mutations), the use of references (eg what reference shows that PTENG130 is lipid and protein phosphatase dead?), and the general flow of the discussion remain inconclusive. For instance, the analysis on the PTEN KI mice and the implication of findings generated in mouse model have not been put in the right context for the review. The review is supposed to compare the role of PTEN lipid and protein phosphatase activity, and the mutations modelled in vivo were generated with this purpose. However, in their analysis, the authors have only discussed the effects of PTEN loss-of-function mutations on various tissues and have not highlighted whether there were differences depending on the loss of PTEN lipid or lipid plus protein phosphatase activity. In addition, the text has not been simplified and the review remains difficult to read.

Overall, I find that in its current status,  the review is not ready for publication and requires additional, extensive work.

Author Response

Reviewer #2 (Comments and Suggestions for Authors):

The accuracy of some paragraphs (please see row 226 to 244, this paragraph is disconnected from the following paragraph even though they discuss the same set of mutations), the use of references (eg what reference shows that PTENG130 is lipid and protein phosphatase dead?), and the general flow of the discussion remain inconclusive. For instance, the analysis on the PTEN KI mice and the implication of findings generated in mouse model have not been put in the right context for the review. The review is supposed to compare the role of PTEN lipid and protein phosphatase activity, and the mutations modelled in vivo were generated with this purpose. However, in their analysis, the authors have only discussed the effects of PTEN loss-of-function mutations on various tissues and have not highlighted whether there were differences depending on the loss of PTEN lipid or lipid plus protein phosphatase activity. In addition, the text has not been simplified and the review remains difficult to read.

We thank the reviewer for these great suggestions and questions. We now updated the references, reorganized this section, and revised the manuscript. In the manuscript, an analysis of PTEN KI mouse studies is used to confirm the function and consequence of PTEN point mutation. We also highlighted the differences in the results due to additional loss of the protein phosphatase activity. Also, due to the limitation of both models without KI of only PTEN protein phosphatase mutant, we cannot conclude the precise function of PTEN protein phosphatase in both animal studies.

Reviewer 3 Report

The Authors have modified the manuscript according to the Reviewers' comments and it is greatly improved. There are still a few errors and text that should be modified; these are detailed below:

Line 50: 3'4-diphosphate should be replaced with "4,5-bisphosphate"

Line 51: Should read "TI loop" and not T1 loop"

Line 72: Replace "losses" with "loses" and "inhibits" with "inhibit"

Line 116: Insert "Mammary epithelial cells from" before "PTEN hypermorphic"

Line 117: Change "80% regular" to "80 of regular"

Lines 126 and 127 right and left panel calls are inverted

Line 129: Change PTN to "PTEN"

Line 138: Add "also" before "known"

Figure 5 - is PDZ really an abbreviation for an acetyltransferase?

Line 330 - change "diver" to "driver"

Line 331 -delete PIP2-3kinase and just write the "PI3K-Akt pathway"

Line 559 - add "act:" after "activity"

Line 561- change "dephosphorylated its substrates" to "dephosphorylation of its substrates"

Line 562 - add "in" between "participate" and "the"

Line 570 - change substrate to "substrates" - network to "networks" - puzzle to "unpuzzle"

Line 571 - change parthenogenic to "pathogenic"

Author Response

Reviewer #3 (Comments and Suggestions for Authors):

The Authors have modified the manuscript according to the Reviewers' comments and it is greatly improved.

There are still a few errors and text that should be modified; these are detailed below:

Line 50: 3'4-diphosphate should be replaced with "4,5-bisphosphate"

Line 51: Should read "TI loop" and not T1 loop"

Line 72: Replace "losses" with "loses" and "inhibits" with "inhibit"

Line 116: Insert "Mammary epithelial cells from" before "PTEN hypermorphic"

Line 117: Change "80% regular" to "80 of regular"

Lines 126 and 127 right and left panel calls are inverted

Line 129: Change PTN to "PTEN"

Line 138: Add "also" before "known"

Figure 5 - is PDZ really an abbreviation for an acetyltransferase?

Line 330 - change "diver" to "driver"

Line 331 -delete PIP2-3kinase and just write the "PI3K-Akt pathway"

Line 559 - add "act:" after "activity"

Line 561- change "dephosphorylated its substrates" to "dephosphorylation of its substrates"

Line 562 - add "in" between "participate" and "the"

Line 570 - change substrate to "substrates" - network to "networks" - puzzle to "unpuzzle"

Line 571 - change parthenogenic to "pathogenic"

The review by Liu et al. provides a general introduction on PTEN function, modes of regulation, covering alternative variants and the different mutants used to pick apart lipid/protein phophatase activities. This is followed by a focus on the less well-known functional effects of PTEN protein phosphatase activity. In general I think it's well constructed and covers most of the important points.

We thank the reviewer for pointing out these errors. We have corrected them in our revised manuscript.